# Supramolecular Self-Assembly Strategy towards Fabricating Mesoporous Nitrogen-Rich Carbon for Efficient Electro-Fenton Degradation of Persistent Organic Pollutants

**DOI:** 10.3390/nano12162821

**Published:** 2022-08-17

**Authors:** Ye Chen, Miao Tian, Xupo Liu

**Affiliations:** School of Materials Science and Engineering, Henan Engineering Research Center of Design and Recycle for Advanced Electrochemical Energy Storage Materials, Henan Normal University, Xinxiang 453007, China

**Keywords:** supramolecule, mesoporous nitrogen-rich carbon, carbon fixation, electro-Fenton, organic pollutant degradation

## Abstract

The electro-Fenton (EF) process is regarded as an efficient and promising sewage disposal technique for sustainable water environment protection. However, current developments in EF are largely restricted by cathode electrocatalysts. Herein, a supramolecular self-assembly strategy is adopted for synthetization, based on melamine–cyanuric acid (MCA) supramolecular aggregates integrated with carbon fixation using 5-aminosalicylic acid and zinc acetylacetonate hydrate. The prepared carbon materials characterize an ordered lamellar microstructure, high specific surface area (595 m^2^ g^−1^), broad mesoporous distribution (4~33 nm) and high N doping (19.62%). Such features result from the intrinsic superiority of hydrogen-bonded MCA supramolecular aggregates via the specific molecular assembly process. Accordingly, noteworthy activity and selectivity of H_2_O_2_ production (~190.0 mg L^−1^ with 2 h) are achieved. Excellent mineralization is declared for optimized carbon material in several organic pollutants, namely, basic fuchsin, chloramphenicol, phenol and several mixed triphenylmethane-type dyestuffs, with total organic carbon removal of 87.5%, 74.8%, 55.7% and 54.2% within 8 h, respectively. This work offers a valuable insight into facilitating the application of supramolecular-derived carbon materials for extensive EF degradation.

## 1. Introduction

As a result of the rapid evolution of industry and agriculture, a huge amount of organic matter, such as dyeing, pharmaceutical and residual organic chemical wastewater and so forth, is discharged into the aquatic ecosystem, inducing serious environmental pollution issues. The electro-Fenton (EF) process, is a widely used electrochemical advanced oxidation process (EAOP) and is accepted as valid in sewage management to comply with measures to protect the water environment. The EF process undergoes a typical two-electron oxygen reduction reaction (2e^−^ ORR) process, which relies heavily on the performance of cathodic catalytic materials [1,2,3]. In terms of appealing cathode catalysts, carbon materials with their advantages of cost-effective availability, eco-friendliness and good electrical conductivity were widely utilized in the EF degradation of various organic contaminants [4,5,6]. However, the current development of EF is hampered by the limited catalytic activity and stability of cathode carbon materials.

Extensive efforts were devoted to improving the performance of carbon materials, including heteroatom doping, functionalization, structural modification and molecular doping (molecular copolymerization), etc. However, common doping or modification of carbon materials have the disadvantages of tedious processing, large consumption of reagents, low doping contents and uncontrolled/non-uniform morphology. Fortunately, simple, feasible and low-toxic molecular copolymerization can eliminate the above shortcomings well. It can induce molecular assembly to generate supramolecular polymers with highly ordered microstructure/macroscopic structures via hydrogen bonding formation [7]; these are beneficial for creating efficient carbon catalysts for EF, following high-temperature treatment.

The self-assembled melamine–cyanuric acid (MCA) is a common supramolecular aggregate obtained by molecular copolymerization and regarded as a promising precursor for preparing high-performance carbon materials [8,9,10,11]. Compared with single nitrogen-containing precursors (such as urea, melamine or dicyandiamide), MCA has the following advantages: (i) MCA can solve, effectively, the problems, such as the small specific surface area and bits of exposed active sites, of carbon materials derived from single precursors. (ii) The supramolecular self-assembly process is beneficial in maintaining a specific ordered morphology at high temperatures due to the non-covalent interactions of hydrogen bonding with strong directionality [12]. Generally, most studies focused on the application of MCA supramolecular aggregates to prepare g-C_3_N_4_ at ~550 °C. Nevertheless, g-C_3_N_4_ with an ultra-high level of nitrogen doping and poor electrical conductivity is far from satisfactory for electrochemical employment. The shortcomings can be well circumvented by further converting g-C_3_N_4_ into N-doped carbon materials with higher conductivity at higher temperature. However, there is little or no residue when the carbonization temperature is further elevated. That is, the carbon yield can be negligible. It was found that the metal-assisted fixation of g-C_3_N_4_-derived carbon is valid in resolving the above issue [13,14,15]. Moreover, suitable metal compounds are not only able to fix the carbon, but they also tailor the pore structures and heteroatom doping of carbon materials, which, in turn, supply sufficient active sites in pursuit of the competent 2e^−^ ORR in the EF reaction [16,17]. Thus, exploring appropriate strategies for utilizing MCA supramolecular aggregates to fabricate efficient cathode carbon catalysts is indispensable for advancing the development of the EF process.

Herein, mesoporous nitrogen-rich carbon materials were established for EF degradation processes by carbonizing MCA supramolecular aggregates. The supramolecular precursor has a significant effect on the microstructure adjustment and N doping of the carbon matrix. The adopted 5-aminosalicylic acid and zinc acetylacetonate hydrate not only play a crucial role in carbon fixation, but also affect the specific surface area and N/O content of carbon materials. The synthesized carbon materials show a regular lamellar-like morphology, large specific surface area, rational structural defect and high nitrogen content, owing to the structural superiority of supramolecular aggregates. The optimized carbon catalysts exhibit a promising application in EF degradation. Numerous organic pollutants, namely, basic fuchsin (BF), mixed triphenylmethane-type dyes, phenolic substances and antibiotics containing chloramphenicol were highly degraded and mineralized with admirable stability and reusability. Therefore, this work supplies a simple approach for the preparation of advanced cathode carbon materials to facilitate the practical applications of EF in organic pollutant treatment.

## 2. Materials and Methods

### 2.1. Chemicals and Materials

Melamine (C_3_H_6_N_6_, MA), cyanuric acid (C_3_H_3_N_3_O_3_, CA), 5-aminosalicylic acid (C_7_H_7_NO_3_), the antibiotic chloramphenicol (C_11_H_12_Cl_2_N_2_O_5_), phenol (C_6_H_6_O), and four triphenylmethane-type dyes of basic fuchsin (C_20_H_20_ClN_3_, BF), malachite green (C_23_H_26_N_2_, MG), victoria blue B (C_33_H_32_ClN_3_, VB) and leucocrystal violet (C_25_H_31_N_3_, LV) were all purchased from Aladdin (Shanghai, China). Zinc acetylacetonate hydrate (C_10_H_14_ZnO_4_·xH_2_O) was supplied by Macklin. Sodium hydroxide (NaOH), sulfuric acid (H_2_SO_4_), ethanol (C_2_H_6_O), hydrochloric acid (HCl), phosphoric acid (H_3_PO_4_), iron (II) sulfate heptahydrate (FeSO_4_·7H_2_O), sodium sulfate (Na_2_SO_4_), isopropyl alcohol (C_3_H_8_O) and potassium titanyl oxalate (K_2_TiO·C_4_O_8_·2H_2_O) were acquired from Alfa Aesar (Shanghai, China). All chemicals were of analytical grade and employed without extra purification.

### 2.2. Preparation of Mesoporous Nitrogen-Rich Carbon Materials

For the typical preparation of carbon materials, 2 g melamine, 2 g cyanuric acid, 0.31 g 5-aminosalicylic acid and different contents of zinc acetylacetonate hydrate were dispersed in 200 mL ethanol under vigorous stirring and subsequent ultrasonic processing for 4 h [10,11]. The milky precipitates were gathered by centrifugation and rinsed with ethanol several times, and then vacuum dried at 60 °C overnight. Afterward, the ground powders were delivered to the tube furnace for calcining at 550 °C (at a ramp rate of 2.3 °C min^−1^), subsequently rising to 900 °C at 5 °C min^−1^ under a continuous N_2_ flow for 2 h. The products were washed in HCl solution and deionized water several times, followed by drying at 40 °C. The finally obtained samples were referred to as MCAN-*x*, in which x represented the mass of zinc acetylacetonate hydrate (*x* = 0.5, 1, 2 and 3 g).

Additionally, the control samples of MAN-1 and CAN-1 were prepared using the same synthetic process except for precipitating the single melamine and cyanuric acid, respectively. The samples without 5-aminosalicylic acid or zinc acetylacetonate hydrate participation were labelled MCA-1 and MCAN-0, respectively. The N-1 sample was also prepared using 5-aminosalicylic acid and zinc acetylacetonate hydrate only.

### 2.3. Characterizations

Field emission scanning electron microscopy (FE-SEM) for the microstructure and morphological characteristics was conducted at Hitachi High-Technologies Corporation SU8010 and allied to transmission electron microscopy (TEM) at JEOL JEM-2100. The textural properties and pore structures were acquired from Quantachrome Autosorb Station iQ2 apparatus. X-Ray Diffraction (XRD) patterns for crystalline structure were employed on the Bruker-D8 apparatus (Cu Kα radiation) in the scope of 5~70°. Raman spectra for the graphitization degree of the samples were measured using a LabRAM HR Evolution (532 nm laser source). The surface element information and functional groups were gathered by the X-ray photoelectron spectroscopy (XPS) spectra (Thermo Scientific ESCALAB250) and Fourier transform infrared spectroscopy (FTIR) spectra (Thermo Nicolet Corporation NEXUS equipment). The hydrophilicity was recorded on a KRüSS DSA25 contact angle meter. A TG/DTA thermal analyzer (NETZSCH STA449F3) was employed for weight variation of the samples.

### 2.4. Electrochemical Measurements

To investigate the electrochemical performance of the obtained catalysts, a cyclic voltammogram (CV) was firstly operated within a three-electrode system (3 mm-diameter glassy carbon coated by catalysts as working electrodes, platinum foil and saturated calomel electrode (SCE) as counter and reference electrodes) and measured by an electrochemical workstation (CHI 660E). The electrolyte was 0.05 mol·L^−1^ Na_2_SO_4_ solution (pH = 2.0) purified with N_2_ or O_2_. Subsequently, the experimental conditions of the linear-sweep voltammogram (LSV) were similar to those of the CV, except that the working electrode diameter was 4 mm. The H_2_O_2_ concentration was estimated by a UV−Vis spectrophotometer (TU-1810) using the titanium (IV) spectrophotometric method at an absorption wavelength of 400 nm.

The EF degradation systems were established for various pollutants and different conditions (cathode potentials with −0.8 V vs. SCE, Fe^2+^ contents, initial concentrations and pH values). The target organic pollutants were BF (C_0_ = 10 mg·L^−1^), the mixed dyes (C_0_ = 10 mg·L^−1^, containing BF, malachite green (MG), victoria blue B (VB) and crystal violet (LV)), the chloramphenicol (C_0_ = 20 mg·L^−1^) and the phenol (C_0_ = 20 mg·L^−1^). O_2_ was supplied for 15 min in the EF systems with the precipitation of FeSO_4_ (0.2 mmol·L^−1^) and Na_2_SO_4_ (0.05 mol·L^−1^). The pH values for all the electrolytes were 2.0 and were adjusted with 1 mol·L^−1^ H_2_SO_4_ and 0.1 mol·L^−1^ NaOH.

The removal efficiency of the pollutant was obtained from Equation (1).
(1)$$\mathrm{removal}\;\mathrm{efficiency}\;(\%)=\frac{C_{0}{-C}_{t}}{C_{0}}\times 100\%$$
where C_t_ and C_0_ represent the final and initial pollutant concentrations (mg L^−1^), respectively.

The pseudo-first-order kinetics were estimated by the following Equation (2).
(2)$$\ln\frac{C_{t}}{C_{0}}=-\mathrm{kt}$$
where the rate constant (k, min^−1^) means the apparent rate constant, the representations of C_t_ and C_0_ are the same as the above.

Furthermore, the mineralization of different pollutants was evaluated by the Vario TOC analyzer (Elementar Analysensysteme GmbH). The mineralization efficiency was obtained by Equation (3).
(3)$$\mathrm{TOC}\;\mathrm{removal}\;(\%)=\frac{{\mathrm{TOC}}_{0}{-\mathrm{TOC}}_{t}}{{\mathrm{TOC}}_{0}}\times 100\%$$
where TOC_0_ and TOC_t_ are the measured TOC values before and after the degradation treatments, respectively.

## 3. Results and Discussion

### 3.1. Catalyst Characterization

Supramolecular self-assembly strategy was proposed to synthesize the mesoporous nitrogen-rich carbon materials. The hydrogen-bonded MCA supramolecular precursor was generated through the molecular cooperative assembly process. MCA supramolecular aggregates were further converted into nitrogen-doped porous carbon after subsequent calcination and pickling treatments with the assistance of the carbon-fixation effect via 5-aminosalicylic acid and zinc acetylacetonate hydrate. The formation mechanism of carbon materials is shown in Figure 1a. MCA supramolecular aggregates are employed as a structure-directing agent for creating a mesoporous lamellar structure. As shown in Figure 1b, the preparative MCA precursor shows the morphology of stacked-sheet aggregates. The morphology becomes much thinner during the calcination process. The obtained MCAN-*x* carbon materials all exhibit a crimped and flexible lamellar-like architecture with uniformly dispersed pores (Figure 1c,d and Appendix A), which is profitable for the exposure of active sites to improve electrocatalytic activity; however, the MAN-1 or CAN-1 carbon materials prepared only with melamine or cyanuric acid show an irregularly blocky structure (Appendix A). The obvious difference in morphologies confirms the noteworthy structural superiority of the supramolecular aggregates via the self-assembly process for the formation of specific flaky carbon nanosheets in contrast with the single precursors. The TEM image of MCAN-1 in Figure 1e,f reveals an appealing nanosheet-assembly structure. The nanosheets possess a lattice space of 0.34 nm (Figure 1g), which corresponds well to the (002) plane of graphitic carbon [18]. The results illustrate that the porous carbon materials with ultrathin graphite nanosheets are successfully synthesized through the supramolecular self-assembly strategy. Porous carbon nanosheets can increase the specific surface areas and expose more active sites, promoting ORR electrocatalytic activity.

To investigate the formation mechanism of porous carbon materials, TG analysis of different precursors was conducted and the results are shown in Figure 2a,b. It is observed that the pure MCA supramolecular aggregates only exhibit a residual mass of 0.9 wt.% after high-temperature calcination. Residuals of only 4.9 wt.% and 1.5 wt.% are achieved for MCAN-0 and MCA-1, respectively. However, an ~10.0 wt.% residual is detected for MCAN-1 at the pyrolysis temperature of 900 °C. It can be inferred that the actively reactive groups of amino, hydroxyl and carboxyl in 5-aminosalicylic acid precipitate in the molecular-assembly process, thereby protecting the carbon skeletons well. The hydroxyl and carbonyl groups interact with the MCA carbon skeleton via hydrogen bonding for carbon matrix fixation. Meanwhile, the decomposition products from zinc acetylacetonate hydrate (such as zinc nitride and zinc oxide) can also prevent the complete decomposition of g-C_3_N_4_ during the calcination process. Together, 5-aminosalicylic acid and zinc acetylacetonate hydrate wield a synergistic impact on the reconstitution process of the gaseous nitrogen-containing carbonaceous species, which transforms the g-C_3_N_4_ intermediates to the MCA-derived N-doped carbon materials. The above carbon-fixation process is in competition with the pyrolysis of the MCA thermal polycondensation-produced g-C_3_N_4_ that both of which volatilize to induce the pore structures. In other words, the MCA precursor not only plays a role in N doping during the evolution process, but also serves as a soft template to create pores and increase the specific surface area by generating the N-containing molecular gas.

In Figure 2c and Appendix A, all the as-synthesized samples exhibit similar XRD patterns. The discernible peak at ~21° and the indistinct peak at ~43° are indexed to (002) and (100) crystal faces of graphitic carbon, respectively [19,20]. There is no metal-related peak observed as the zinc doping amounts gradually increase, signifying that the metal species are thoroughly removed. It is notable that the XRD peaks of MAN-1 and CAN-1 at ~23° exhibit a positive shift compared with the carbon materials derived from the MCA supramolecular precursor, revealing more defects and broader interlayer distances for MCAN-*x* [14,21]. Furthermore, the defect density was explored by Raman spectra. In Figure 2d and Appendix A, the characteristic peaks at ~1347 cm^−1^ (D band) and ~1560 cm^−1^ (G band) point to the disordered carbon and sp^2^-hybridized carbon, respectively [22,23]. The *I*_D_/*I*_G_ values of all samples are in the range of 1.38~1.53 with massive defective sites. The defects originating from the N-doping and porous structures can efficiently promote the H_2_O_2_ transfer process from the cathode surface and further accumulate the pivotal OOH* intermediates to hinder the generation of H_2_O in electro-Fenton degradation [24].

The surface C, N and O contents of MCAN-1 are estimated as 75.99 at.%, 19.62 at.% and 4.39 at.%, respectively, by XPS measurement (listed in Table 1), showing an obvious carbon-fixation effect from the self-assembly strategy based on 5-aminosalicylic acid and zinc acetylacetonate hydrate. The related hydrophilic groups contribute to a strong hydrophilicity with a contact angle of 5.5° for MCAN-1 (Figure 2e inset) [25]. The N-rich feature of catalysts causes a favorable electron-donating ability, which enhances electron delocalization and, consequently, creates sufficient active sites for ORR [26,27]. The MCAN-*x* samples are enriched with N functional groups, containing pyridinic N (~398.3 eV), pyrrolic N (~399.8 eV), graphitic N (~401.1 eV) and oxidized N (~403.8 eV) (Figure 2e) [28]. The deconvoluted O 1s spectra exhibit four peaks related to C=O (~531.3 eV), C–OH (~532.3 eV), C–O–C (~533.5 eV) and O–C=O (~534.5 eV) bonds (Figure 2f). The latter two groups are widely admitted as active sites to facilitate the reduction process from O_2_ to H_2_O_2_ [29,30]. The N contents in the carbon matrix can be readily tailored via varying different precursors, such as MAN-1 (15.61 at.%), CAN-1 (13.6 at.%) and especially N-1 (without MCA precursor, 5.79 at.%), indicating that the self-assembly of MCA provides a large amount of nitrogen (Table 1 and Appendix A). Differing from the MCAN-0 (13.42 at.%), the N contents of the samples with Zn salts are 15.41~19.62 at.%. Thus, it can be inferred that the partial N from MCA supramolecular aggregates can be retained in the carbon frameworks during the carbon-fixation process by Zn species, while the remaining MCA decomposes resulting in pore creation during high-temperature calcination.

Specific surface area and porosity were explored by N_2_ adsorption–desorption experiments. All the samples feature similar Type IV isotherms with H3 hysteresis loops (Figure 2g and Appendix A), which correspond to widespread slit-shaped mesoporosity in pore size distributions (PSDs) (Figure 2h,i and Appendix A). The specific surface areas of MCAN-*x* are located in the range of 374~626 m^2^·g^−1^ with pore volumes of 2~4 cm^3^·g^−1^. As expected, the MCAN-*x* samples derived from MCA supramolecular aggregates exhibit much larger surface areas than those of MAN-1 (84 m^2^·g^−1^) and CAN-1 (77 m^2^ g^−1^), demonstrating that the textural characteristics of carbon materials can be improved by the supramolecular self-assembly strategy [31]. In addition, the lower specific surface area of N-1 attains 467 m^2^ g^−1^ compared with that of MCAN-1 with the addition of MCA precursor (595 m^2^ g^−1^), which further attests to the pore-forming ability of MCA. The results match well with the former SEM/TEM analysis. The higher surface area can expose more active species and provide desirable access to the reactants and the cathode material surface, thereby facilitating, in a promising way, the 2e^−^ ORR catalytic activity. In addition, similar PSDs are acquired and distributed at about 4~33 nm for the synthesized samples. The widespread mesoporous structures can act as diffusion channels for the O_2_ and penetration of electrolytes, consequently expediting the activity and selectivity of H_2_O_2_ generation via the 2e^−^ ORR pathway.

### 3.2. Electrochemical Measurements

To unravel ORR activity and selectivity in the representative sample of MCAN-1, CV and LSV measurements were conducted in both N_2_/O_2_-saturated Na_2_SO_4_ electrolytes (pH = 2.0). Upon O_2_ saturating, MCAN-1 exhibits a distinct reduction peak at −0.064 V (vs. SCE), as shown in Figure 3a. In view of the diffusion-controlled measurements, the limiting current density gradually rises with the increased rotation rates (400–2025 rpm), as shown in Figure 3b. The corresponding Koutecky–Levich (K–L) curves are further estimated and feature a definite linearity (inset of Figure 3b). The electron transfer numbers (*n*) are calculated to be about 2.0~2.6 in the potential range of −0.8~−0.3 V from the K–L equation, indicating a two-electron ORR process (Figure 3c). Besides, the uninterrupted H_2_O_2_ accumulation in MCAN-1 was recorded to be ~190.0 mg·L^−1^ with 120 min, which is much higher than most of the reported H_2_O_2_ yields (Figure 3d) [32,33,34,35,36,37]. The depletion of H_2_O_2_ concentrations can be ignored after three cycles of continuous measurement by adopting the same MCAN-1 cathode, indicating its noteworthy durability. The H_2_O_2_ can be activated into •OH on cathode surfaces and bulk solution for further mineralization of the organic dye molecules to CO_2_ and H_2_O [38].

### 3.3. Degradation of Organic Pollutants

The application of synthetic samples in the EF degradation of organic pollutants was further investigated. The schematic diagram of the EF degradation process is depicted in Figure 4a. The removal efficiency (C/C_0_) of BF is employed to evaluate the performances of MCAN-x and control samples with −0.8 V. As shown in Figure 4b and Appendix A, the C/C_0_ value of MCAN-1 after 60 min was determined to be 0.021, which is the lowest among all the samples, demonstrating the superior degradation ability. The excellent performance is attributed to the large S_BET_ and porous characteristic for accelerating O_2_ diffusion and providing plentiful active sites for H_2_O_2_ generation. The chemical kinetics of the oxidative degradation process was further investigated. As Figure 4c, Appendix A and Table 1 show, the −ln(C/C_0_) has a linear relationship to the reaction time (t) for all samples, indicating the typical pseudo-first-order kinetics of the BF degradation reaction. The k from the curve slope of −ln(C/C_0_) vs. t directly reflects the degradation rate of BF, in which MCAN-1 with the highest k-value of 0.063 min^−1^ has the fastest degradation reaction process. The degradation efficacy of MCAN-1 can reach 98.0% in the short period of 1 h, evaluated by UV−Vis curves (Figure 4d). The absorption peaks of BF dyestuff are intuitively faded away accompanied by the magenta decoloration (Figure 4e,f). Thus, the as-prepared MCAN-1 delivers promising application in EF degradation of organic pollutants.

In addition, isopropanol generally acts as an •OH scavenger and was added to the BF solution for the purpose of demonstrating the contribution of •OH during EF degradation. As a result, the BF degradation efficiency is apparently inhibited at ~51.8% (k = 0.011 min^−1^) within 1 h (Figure 5a), confirming the significance of •OH species on the oxidation degradation of BF dyestuff in the EF process. To investigate the practicability of catalysts, their electrochemical stability and reusability for the oxidization of organic contaminants were further explored. The response of the current shows that they do not decay alongside the continuous mineralization process of BF within ~8 h (Figure 5b). The relatively steady current manifests a good stability in MCAN-1, which profits from the stable carbon skeletons and porous features for active site preservation. This also implies an indirect cathodic oxidation process (EF) instead of the direct electrochemical destruction during the dyestuff removal [39]. Furthermore, the MCAN-1 catalyst only yields about 6% removal efficiency recession (~98% to ~92%) within 10 successive runs (Figure 5c, Appendix A and Appendix A) because of its well-preserved reusability and stability during the EF degradation process.

To achieve the optimized degradation conditions, the influencing parameters for EF degradation of BF were also systematically exploited for the MCAN-1 catalyst, including Fe^2+^ contents, pH values, initial concentrations and cathode potentials. Firstly, the influence of Fe^2+^ contents was investigated as shown in Figure 5d (top). Poor degradation efficiency results from a low concentration of 0.1 mM being supplied. An enhanced degradation efficiency is seen with the incremental Fe^2+^ content (0.2 mM), which supplies sufficient •OH species and strong oxidation ability [40]. However, as the concentrations increase further, a negative BF-oxidation efficiency is presented resulting from the •OH depletion by the redundant Fe^2+^ [24]. The k-values are calculated to be 0.026, 0.063, 0.043 and 0.030 min^−1^ from 0.1 mM to 0.8 mM (Appendix A and Appendix A). The highest k-value was obtained for 0.2 mM, indicating the fastest reaction mechanism. Thus, 0.2 mM is selected as the optimal Fe^2+^ concentration for the EF degradation of BF. Secondly, it is well known that the pH values of electrolytes have an important effect on ORR properties, owing to the proton-coupled electron transfer process [41]. The BF degradation efficiencies were explored in electrolytes with various pH values, with results shown in Figure 5d (bottom). Maximum efficiency was achieved at pH = 2.0. A distinct decrement in the BF degradation ability from 98.0% to 22.3% was obtained with the relevant k values from 0.063 min^−1^ to 0.004 min^−1^ when the pH values rose from 2.0 to 7.0 (Appendix A and Appendix A). The decreased acidity causes H_2_O_2_ decomposition and insoluble iron precipitation, thereby reducing catalytic activity and hindering sustainably oxidative degradation performance [24,42].

Furthermore, the impact of initial BF concentrations on degradation performance was concretely explored in Figure 5e (top). As anticipated, intensive BF degradation efficiency is accomplished at 5 mg L^−1^ (the lowest concentration) due to the fact that the few dyestuff molecules must be degraded based on the interfacial reaction at the same •OH production ability. All the removal efficiencies decrease as the initial concentrations gradually increase. The decreasing *k* values of 0.079, 0.063, 0.026, 0.018, 0.012 and 0.008 min^−1^ are presented from 5 to 50 mg·L^−1^ initial BF concentrations, respectively (Appendix A and Appendix A). In addition, the effect of supplied potentials was studied. In general, the supplied potentials manage the electron transfer between two electrodes and have a significant effect on H_2_O_2_ production. Furthermore, the appropriate potential is also beneficial in reducing the energy consumption [24]. In Figure 5e (bottom), the different potentials are compared against BF degradation using the same MCAN-1 cathode. As the applied potentials increase from −0.2 V to −1.0 V, the degradation efficiency gradually rises and then declines, with the optimum performance at −0.8 V. The negative potentials are conducive to the high-efficiency electron transfer, promoting the reduction of O_2_ to H_2_O_2_, while excessive potentials lead to the preferential H_2_ evolution or H_2_O_2_ degradation with electron consumption, consequently reducing the current efficiency [34,43]. In Appendix A and Appendix A, the degradation process follows the pseudo-first-order kinetics at different potentials. The corresponding k-value of MCAN-1 under −0.8 V is also higher than those of −0.2 V (0.017 min^−1^), −0.4 V (0.028 min^−1^), −0.6 V (0.039 min^−1^) and −1.0 V (0.021 min^−1^), indicating that the fastest degradation rate can be reached when the potential is −0.8 V. As a result, the optimal efficiency for EF degradation based on MCAN-1 is achieved in Na_2_SO_4_ electrolyte (appropriate concentration, 0.2 mM of Fe^2+^, pH = 2.0) at the reaction voltage of −0.8 V (Figure 5f). The as-prepared MCAN-1 catalyst shows a potential application for practical contaminant removal. Furthermore, the mineralization performances of all the prepared catalysts were assessed by TOC measurement. As can be seen from Figure 5g and Table 1, MCAN-1 exhibits the optimal mineralization efficiency with a TOC removal value of 87.5%, much higher than the efficiencies of MCAN-0.5 (81.8%), MCAN-2 (73.3%), MCAN-3 (70.3%), MCA-1 (57.3%), N-1 (57.0%), MCAN-0 (52.5%), MAN-1 (49.9%) and CAN-1 (43.6%). The consistency of the EF degradation capacity for the MCAN-1 cathode was also studied with the antibiotic chloramphenicol, phenol, and the mixed triphenylmethane-type dyes containing BF, MG, VB and CV as representative contaminants. The TOC removal efficiencies of MCAN-1 were determined to be 74.8%, 55.7% and 54.2% within 8 h for the above three target contaminants (Figure 5h), revealing its appealing degradation ability for various types of organic contaminants.

## 4. Conclusions

In summary, mesoporous nitrogen-rich carbon materials were fabricated relying on the MCA supramolecular aggregates with the assistance of 5-aminosalicylic acid and zinc acetylacetonate hydrate. The supramolecular precursor has a significant effect on the microstructure of the carbon material, owing to the non-covalent interactions of hydrogen bonding with strong directionality. The optimized MCAN-1 is composed of a lamellar morphology, high N content of 19.62 at.%, abundant mesoporosity and a large specific surface area, resulting in enhanced exposure of active sites for facilitating 2e^−^ ORR. The as-prepared carbon materials exhibit noteworthy potential application in the degradation and mineralization of different models of organic pollutants in EF systems. Serving as the cathode for EF degradation, the representative MCAN-1 delivers excellent mineralization ability for basic fuchsin (87.5%), mixed triphenylmethane-type dyes (54.2%), phenol (55.7%) and chloramphenicol (74.8%) in 8 h. The carbon materials containing desirable stability and reusability show a promising application prospect for contaminant removal. This work provides a simple and delicate approach to synthesizing the mesoporous N-rich carbon materials with adjustable microstructures to be used as promising electrocatalysts for aquatic environmental pollution improvement.

## Figures and Tables

**Figure 1 nanomaterials-12-02821-f001:**
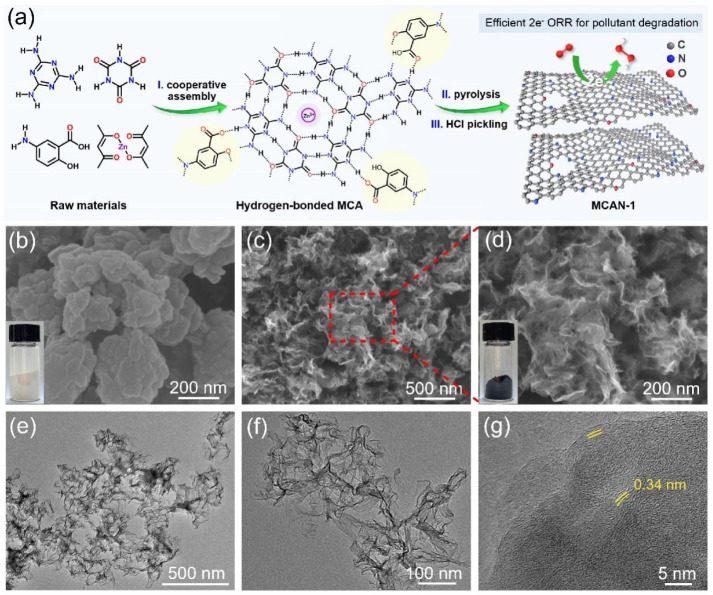
(**a**) MCAN-1 synthetic procedure; (**b**) SEM images of the MCAN-1 precursor (inset: the corresponding photograph); (**c**,**d**) SEM images with different magnifications of the obtained MCAN-1 product (inset: the corresponding photograph of MCAN-1); (**e**,**f**) TEM and (**g**) HR-TEM MCAN-1 images.

**Figure 2 nanomaterials-12-02821-f002:**
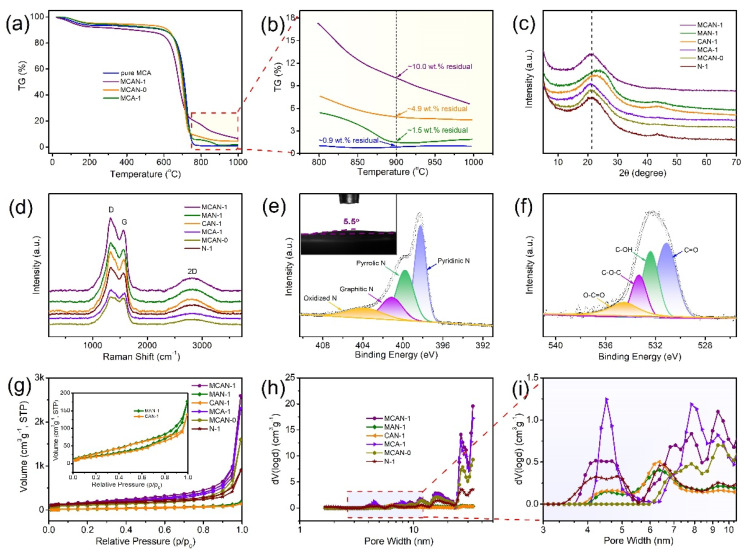
(**a**) Thermogravimetric analysis of intermediate products of pure MCA, MCAN-1, MCAN-0 and MCA-1; (**b**) enlarged figure of thermogravimetric curves; (**c**) XRD patterns and (**d**) Raman spectra for MCAN-1, MAN-1, CAN-1, MCA-1, MCAN-0 and N-1. High-resolution XPS spectra for (**e**) N 1s (inset is contact angle of representative MCAN-1) and (**f**) O 1s for MCAN-1; (**g**) N_2_ adsorption/desorption isotherm (illustration: enlarged isotherms for MAN-1 and CAN-1); (**h**) pore size distribution for MCAN-1, MAN-1, CAN-1, MCA-1, MCAN-0 and N-1 and (**i**) enlarged figure of pore size distributions at about 3~10 nm.

**Figure 3 nanomaterials-12-02821-f003:**
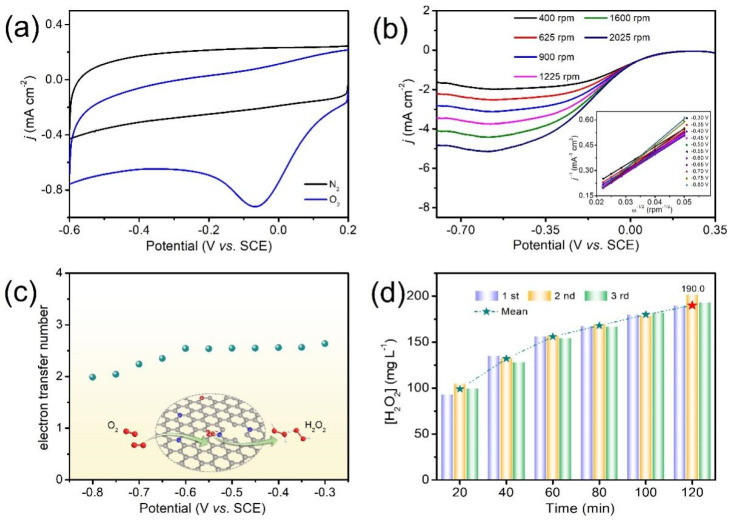
(**a**) CV curves and (**b**) RDE curves at 400~2025 rpm (illustration: the corresponding K–L plot) of MCAN-1; (**c**) electron transfer numbers acquired at diverse potentials of MCAN-1; (**d**) accumulated H_2_O_2_ yields.

**Figure 4 nanomaterials-12-02821-f004:**
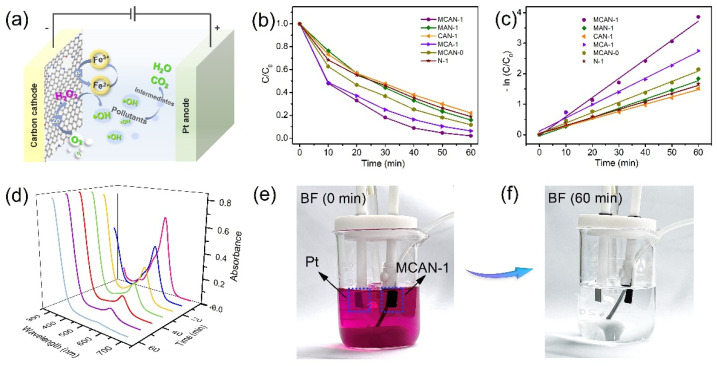
(**a**) Schematic diagram of EF degradation; (**b**) EF degradation of BF by MCAN-1 and the control samples and (**c**) the corresponding variation of −ln(C/C_0_). (**d**) UV−Vis spectra of BF degradation for MCAN-1. Photographs of color changes for basic fuchsin solutions (**e**) before and (**f**) after EF degradation process.

**Figure 5 nanomaterials-12-02821-f005:**
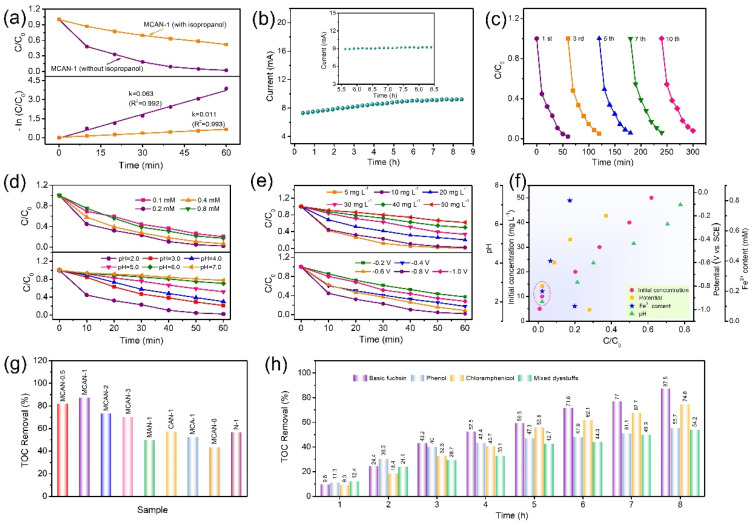
(**a**) The effect of radical scavenger (isopropanol); (**b**) chronoamperometry curve for MCAN-1 (illustration: enlarged curve at about 5.5~8 h); (**c**) reusability of MCAN-1 for BF degradation; (**d**) EF degradation for BF using MCAN-1 cathode with different Fe^2+^ contents (**top**) and pH values (**bottom**); (**e**) EF degradation for BF by using MCAN-1 cathode with different initial concentrations (**top**) and potentials (**bottom**); (**f**) analysis of the optimized degradation conditions for BF degradation process; (**g**) TOC removal of BF for all samples; (**h**) TOC removal of BF, phenol, chloramphenicol and mixed triphenylmethane-type dyes by MCAN-1.

**Table 1 nanomaterials-12-02821-t001:** Summary of characteristics and performance for the synthesized samples.

**Sample**	^ **(A)** ^ **S_BET_** **[m^2^·g^−1^]**	***I*_D_/*I*_G_**	**N Content** **[at.%]**	**O Content** **[at.%]**	^ **(B)** ^ **Removal Efficiency [%]**	***k* [min^−1^]** **(R^2^)**	^ **(C)** ^ **TOC Removal [%]**
MCAN-0.5	626	1.47	15.42	8.66	95.7	0.050 (0.992)	81.8
MCAN-1	595	1.52	19.62	4.39	98.0	0.063 (0.992)	87.5
MCAN-2	492	1.51	19.34	4.94	93.2	0.044 (0.990)	73.3
MCAN-3	397	1.53	18.26	5.12	90.9	0.039 (0.993)	70.3
MAN-1	84	1.44	15.61	6.21	84.1	0.030 (0.995)	49.9
CAN-1	77	1.41	13.63	7.25	78.0	0.024 (0.994)	43.6
MCAN-0	374	1.38	13.42	4.03	88.3	0.034 (0.993)	52.5
MCA-1	494	1.43	14.39	4.24	93.6	0.043 (0.990)	57.3
N-1	467	1.44	5.79	10.38	81.1	0.026 (0.990)	57.0

^(A)^ Specific surface area of the samples; ^(B)^ BF removal efficiency for different samples within 1 h; ^(C)^ TOC removal in EF degradation of BF dyestuff for different samples after 8 h.

## Data Availability

The data presented in this study are available on request from the corresponding author.

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
