# Peer review of "Supramolecular Self-Assembly Strategy towards Fabricating Mesoporous Nitrogen-Rich Carbon for Efficient Electro-Fenton Degradation of Persistent Organic Pollutants"

_nanomaterials, 2022, doi:10.3390/nano12162821_

Round 1

Reviewer 1 Report

Comments and suggestions are specified in the following points.

Scale bars in figures 1b and 1c are 200 nm and 500 nm, respectively, but figure 1c is apparently a view at higher magnification. According to the caption, the same material is reported in both pictures. Scale bars or caption should be checked.

Values of voltage and current (or current density) are not reported for EF degradation experiments (lines 140-147 and 290-317), except in figure 5b. It should be interesting to report these data for EF degradation tests whose results are reported in figures 4b and 4c.

Editing instructions are reported in place of caption to table 1 (line 388)

Author Response

Comment 1: Scale bars in figures 1b and 1c are 200 nm and 500 nm, respectively, but figure 1c is apparently a view at higher magnification. According to the caption, the same material is reported in both pictures. Scale bars or caption should be checked.

Response: Thank you very much for your kind suggestion. The caption of Figure 1 has revised as follows: Figure 1. (a) The synthetic procedure of MCAN-1. (b) SEM images of the precursor of MCAN-1 (inset: the corresponding photograph). (c-d) SEM images with different magnifications of the obtained product of MCAN-1 (inset: the corresponding photograph of MCAN-1). (e-f) TEM and (g) HR-TEM images of MCAN-1.

Comment 2: Values of voltage and current (or current density) are not reported for EF degradation experiments (lines 140-147 and 290-317), except in figure 5b. It should be interesting to report these data for EF degradation tests whose results are reported in figures 4b and 4c.

Response: Thank you very much for your kind suggestion. The EF processes were conducted with the constant voltage. Thus, the voltages were added in line 141 with “cathode potentials with -0.8 V vs. SCE”, and in line 296 with “MCAN-x and control samples with -0.8 V”. All the added contents were marked in gray highlight in the revised manuscript.

Comment 3: Editing instructions are reported in place of caption to table 1 (line 388)

Response: Thank you very much for your kind suggestion. The instructions of table 1 were edited (line 393).

Reviewer 2 Report

In this work,  Supramolecular self-assembly strategy towards fabricating mesoporous nitrogen-rich carbon for efficient electro-Fenton degradation of persistent organic pollutants. The idea of this research is  interesting to readers. The background is well studied and the presentation of the  method is very clear and sound, but there are some minor issues to be addressed:

The author should provide suitable reference in the section 2.2. Preparation of mesoporous nitrogen-rich carbon materials.

In the results and discussion sections, the author need to be provide sub division in each studies. Ex 3.1. FESEM 3.2. TGA, and 3.3. XPS analysis etc.,

The author should discuss in details of the mechanism of synthesize based on melamine-cyanuric acid integrated with carbon fixation of aminosalicylic acid and zinc acetylacetonate hydrate.

Author Response

Comment 1: The author should provide suitable reference in the section 2.2. Preparation of mesoporous nitrogen-rich carbon materials.

Response: Thank you very much for your kind suggestion. The suitable references were added in the section “2.2. Preparation of mesoporous nitrogen-rich carbon materials” of the revised manuscript.

Comment 2: In the results and discussion sections, the author need to be provide sub division in each studies. Ex 3.1. FESEM 3.2. TGA, and 3.3. XPS analysis etc.

Response: Thank you very much for your kind suggestion. According to your suggestion, the logic structure and modular division were presented in the Results and discussion section of the revised manuscript.

Comment 3: The author should discuss in details of the mechanism of synthesize based on melamine-cyanuric acid integrated with carbon fixation of aminosalicylic acid and zinc acetylacetonate hydrate.

Response: Thank you very much for your kind suggestion. The details were added in line 195 as “The hydroxyl and carbonyl groups can interact with the carbon skeleton of MCA via the hydrogen bonding for fixing the carbon matrix.”
